# SpatialRank: Urban Event Ranking with NDCG Optimization on Spatiotemporal Data

**Bang An**
Department of Business Analytics
University of Iowa
Iowa City, IA 52242
`bang-an@uiowa.edu`

**Xun Zhou** *
Department of Business Analytics
University of Iowa
Iowa City, IA 52242
`xun-zhou@uiowa.edu`

**Yongjian Zhong**
Department of Computer Science
University of Iowa
Iowa City, IA 52242
`yongjian-zhong@uiowa.edu`

**Tianbao Yang**
Department of Computer Science and Engineering
Texas A&M University
College Station, TX 77843
`tianbao-yang@tamu.edu`

## Abstract

The problem of urban event ranking aims at predicting the top-$k$ most risky locations of future events such as traffic accidents and crimes. This problem is of fundamental importance to public safety and urban administration especially when limited resources are available. The problem is, however, challenging due to complex and dynamic spatio-temporal correlations between locations, uneven distribution of urban events in space, and the difficulty to correctly rank nearby locations with similar features. Prior works on event forecasting mostly aim at accurately predicting the actual risk score or counts of events for all the locations. Rankings obtained as such usually have low quality due to prediction errors. Learning-to-rank methods directly optimize measures such as Normalized Discounted Cumulative Gain (NDCG), but cannot handle the spatiotemporal autocorrelation existing among locations. In this paper, we bridge the gap by proposing a novel spatial event ranking approach named SpatialRank. SpatialRank features adaptive graph convolution layers that dynamically learn the spatiotemporal dependencies across locations from data. In addition, the model optimizes through surrogates a hybrid NDCG loss with a spatial component to better rank neighboring spatial locations. We design an importance-sampling with a spatial filtering algorithm to effectively evaluate the loss during training. Comprehensive experiments on three real-world datasets demonstrate that SpatialRank can effectively identify the top riskiest locations of crimes and traffic accidents and outperform state-of-the-art methods in terms of NDCG by up to $12.7\%$.

## 1 Introduction

Given a risk score defined based on historical urban events (e.g., crimes, traffic accidents) in a study area as well as the socio-environmental attributes (e.g., travel demands, weather, road network) associated with the events, the goal of the *urban event ranking problem* is to learn a model that can predict the ranking of the top-$k$ riskiest locations of future events.

---

* corresponding author

37th Conference on Neural Information Processing Systems (NeurIPS 2023).

The urban event ranking problem is a widely observed spatiotemporal prediction problem, which has critical applications in public safety, traffic management, and urban planning. For example, the National Highway Traffic Safety Administration [27], estimated a total of 42,939 deaths in motor vehicle traffic crashes in the United States in 2021. Crime control and property loss cost over $2.5 trillion in the United States in 2017 [25]. Chicago Police Department has utilized criminal intelligence analysis and data science techniques to help command staff determine where best to deploy resources [1]. However, according to the Police Executive Research Forum, there were $42.7\%$ more resignations among law enforcement but a $3.9\%$ decrease in hiring new officers in 2021 compared to 2019 [3]. Meanwhile, the Federal Bureau of Investigation confirms that violent crime in 2020 has surged nearly $30\%$ over 2019 [2]. Therefore, given such growth of crimes and accidents, predicting the riskiest locations of traffic accidents or crimes helps law enforcement stakeholders allocate their limited resources strategically to reduce injuries, deaths, and property losses.

Despite its importance, the urban event ranking problem is technically challenging. **First**, there exist complex and dynamic spatiotemporal correlations between locations in terms of events and attributes such as traffic conditions and weather changes. Capturing such dynamic relationships is non-trivial. **Second**, due to the presence of spatial autocorrelation, nearby locations may share very similar socio-environmental attributes. It is thus very challenging to learn a good model that can correctly rank neighboring locations. **Finally**, urban events are usually sparse in space. Making a prediction with a low average error in count does not ensure a good ranking of the top-$k$ locations.

Prior works on urban event prediction employed either classic machine learning methods [15] [4] [7] [9] [11] [12] [18] or deep learning techniques including recurrent neural networks [30] and convolutional neural networks [26][36]. Recently, Li et al. [20] proposed a cross-region hypergraph structure network to address the data sparsity issues. Besides, Yuan et al. [40] [5] proposed spatial ensembles to address spatial heterogeneity. The objective of the above methods is to make accurate predictions of the risk or count for all locations. However, urban events are often very sparse in space, which might guide the model to avoid predicting accidents in most locations to obtain a low average error. Such prediction does not truly benefit the users such as police officers. The top-$k$ locations derived from such predictions are naturally inaccurate.

Our problem is also relevant to the learning-to-rank problem frequently studied in the field of recommendation systems. State-of-the-art methods in this area typically predict the top-$k$ items most likely to be chosen by users through optimizing ranking-based metrics such as the Normalized Discounted Cumulative Gain (NDCG) [8]. The rank operator of NDCG is non-differentiable, and previous studies have made noticeable progress in approximating NDCG by surrogate functions [32] [28] [34] [29]. In our problem, a straightforward adaptation would be to consider the locations as "items" and each time slot as a "user". However, NDCG is not a perfect objective function for our problems as it neither measures the local ranking quality nor considers spatial autocorrelation among locations. Instead, the existing approaches assume items are independent. Therefore, directly applying existing NDCG optimization solutions might not be the best solution to our problem.

In this paper, we bridge the gaps in both fields by formulating the urban event ranking problem as a spatial learning-to-rank problem and solving it by directly optimizing a "spatial" version of the NDCG measure. We propose **SpatialRank**, our deep learning model with three novel designs. To efficiently capture the spatial and temporal dependencies, we design an adaptive graph convolution layer that learns the correlations between locations dynamically from features and historical events patterns; to ensure the model balances both the global ranking quality on all the locations and the local ranking quality on subsets of locations, we propose a hybrid loss function combining both NDCG loss and a novel local NDCG loss; to improve the effectiveness of optimizing NDCG surrogates in the spatiotemporal setting of our problem, we design an importance-based location sampling with spatial filtering algorithm to iteratively adjust the weights of each location considered in the objective function, thereby guiding the model to concentrate on learning for more important locations. We conduct comprehensive experiments on three real-world datasets collected from Chicago and the state of Iowa. The results demonstrate that SpatialRank can substantially outperform baselines and achieve better ranking quality.

Our contributions are summarized below:
- To the best of our knowledge, this is the first paper to formulate urban event forecasting as a location ranking problem and learn a ranking model by optimizing its ranking quality.
- We propose SpatialRank with adaptive graph convolution layers learning from historical event patterns and spatiotemporal features to capture dynamic correlations over time and space.

- We propose a hybrid objective function to leverage the trade-off between global ranking quality and local ranking quality.
- We propose a ranking-based importance sampling algorithm to adaptively adjust the weights of different locations considered in the objective function of the prediction results from the last training epoch to help the model focus on important locations.

## 2 Preliminaries

### 2.1 Formulation of the event ranking problem

A spatio-temporal field $\mathcal{S} \times T$ is a three-dimensional partitioned space, where $T = \{t_1, t_2, ..., t_T\}$ is a study period divided into equal-length intervals (e.g., hours, days) and $\mathcal{S} = \{s_{(0,0)}, ..., s_{(M,N)}\}$ is a $M \times N$ two-dimension spatial grid partitioned from the study area. A set of socio-environmental features $F$ are observed over $\mathcal{S} \times T$, which include temporal features $F_t \in \mathbb{R}^T$ (e.g., day of week, holiday), spatial features $F_s \in \mathbb{R}^{M \times N}$ (e.g., total road length, speed limit), and spatiotemporal features $F_{st} \in \mathbb{R}^{M \times N \times T}$ (e.g., traffic volume, rainfall amount). A risk score $y \in \mathbb{R}^{M \times N \times T}$ is a user-defined attribute over $\mathcal{S} \times T$ measuring the risk level of each spatiotemporal location (e.g., number of crimes, injuries of traffic accidents). $y(s, t) > 0$ when any events occurred in $(s, t)$, and equals 0 when no events occurred.

Given the socio-environmental features $F$ and risk score $y$ for all the locations in time window $\{t_1, t_2, .., t_n\}$, the urban event ranking problem is to predict the ranking of the top-$k$ locations $\{s_1, ..., s_K\} \in S$ in the next time interval $t_{n+1}$ with the highest risk scores. The objective is to prioritize the ranking quality on the top-$k$ riskiest locations. As a basic assumption of our problem, there exists spatial and temporal autocorrelation among locations in $F$, meaning nearby locations tend to have more correlated values. In addition, events are sparse so $y$ is 0 for the majority of the locations. A detailed example of data attributes and feature generation steps can be found in the supplementary materials Appendix A.

### 2.2 Stochastic Optimization of NDCG in Event Ranking Problem

In the field of recommendation systems, learning to rank is substantially studied, and NDCG is widely used as the metric to measure the ranking quality of the foremost importance. In the following part, we use the terminologies from the event forecasting problem to define NDCG in our problem setting. For a ranked list of locations $s \in \mathcal{S}$ in a period $t \in T$, the NDCG score is computed as by:

$$\text{NDCG}_t = \frac{1}{Z_t} \sum_{s \in \mathcal{S}_t} \frac{2^{y_s} - 1}{\log_2(1 + \text{r}(s))}, \tag{1}$$

where $y_s$ is the risk score of the location $s$, $\text{r}(s)$ denotes the rank of location $s$ in the studied spatial domain $\mathcal{S}_t$, and $Z_t$ is the Discounted Cumulative Gain (DCG) score [17] of the perfect ranking of locations for time period $t$. However, the rank operator in NDCG is non-differentiable in terms of model parameters, and thus cannot be optimized directly. A popular solution is to approximate the rank operator with smooth functions and then optimize its surrogates [34][28][29] as shown in Eq. 2.

$$\bar{g}(\mathbf{w}; \mathbf{x}, \mathcal{S}_t) = \sum_{s' \in \mathcal{S}_t} \ell(h_t(s'; \mathbf{w}) - h_t(\mathbf{s}; \mathbf{w})), \tag{2}$$

where rank operator $r(s)$ in NDCG is approximated by a differentiable surrogate function $\ell()$, and the squared hinge loss $\ell(x) = \max(0, x + c)^2$ is commonly used [38]. In this way, the model parameters $\mathbf{w}$ can be updated by a gradient-based optimizer. We can maximize over $L(\mathbf{w})$:

$$\max_{\mathbf{w} \in \mathbb{R}^d} L(\mathbf{w}) := \frac{1}{|T|} \sum_{t=1}^{T} \sum_{s \in \mathcal{S}_t^+} \frac{2^{y_s^t} - 1}{Z_t \log_2(\bar{g}(\mathbf{w}; \mathbf{x}_s^t, \mathcal{S}_t) + 1)}. \tag{3}$$

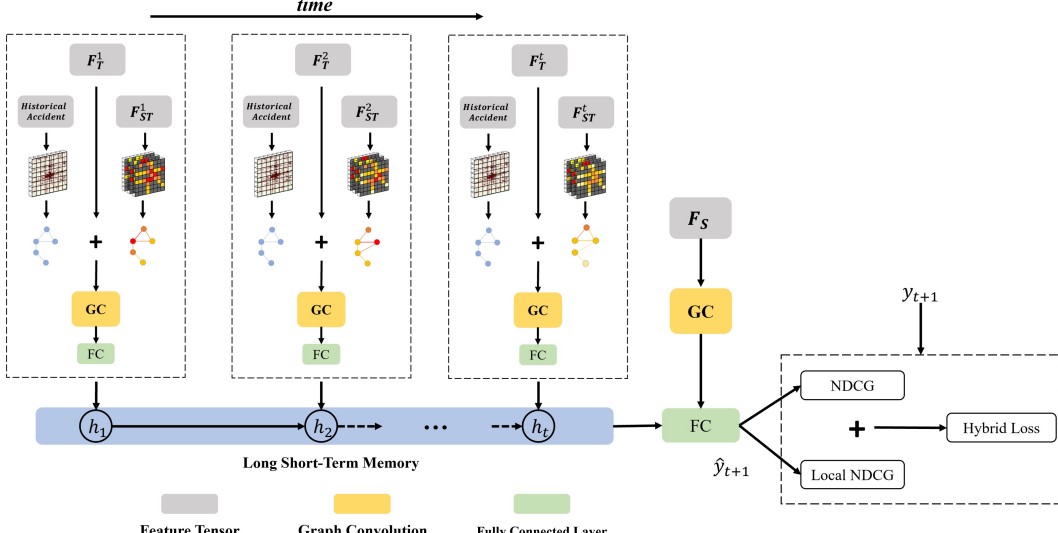

Figure 1: SpatialRank Architecture. The spatiotemporal features are used to generate adjacency matrices and then embedded by graph convolution layers. We use a fully connected layer to make final predictions. The hybrid objective function is combined with NDCG loss and local NDCG loss.

where $\mathbf{s}^t \in S_t^+$ denotes a set of locations with positive risk scores to be considered in the objective function. In this way, the optimization solution on NDCG can be directly applied to the event ranking problem.

# 3 Methodology

Directly optimizing NDCG as described above might provide a solution to our problem but fails to capture the spatiotemporal autocorrelation in the data. Also, since NDCG is a global measure, the model might not be able to learn how to rank nearby locations with highly correlated features correctly. In this section, we present our SpatialRank method to address these limitations. Figure. 1 demonstrates the proposed model architecture.

## 3.1 The Deep Learning Model

Many recent deep learning models for spatiotemporal could be used as the backbone architecture of our SpatialRank method. A key idea in these methods is to model the correlations between locations using a graph, with locations as nodes and edge weights representing the strength of correlations, and extract latent spatial dependency information through graph convolution layers [37] [5]. We follow a similar idea to build the overall network architecture, where spatial and spatiotemporal features are fed into graph convolutional layers. Then the extracted latent representations are concatenated with the temporal features and fed into LSTM layers to capture temporal representations before the final output layer.

A key novelty in our SpatialRank network is the design of a time-guided and traffic-aware graph-generating process for the graph convolution layers. Prior works typically use Pearson's correlation coefficients or similar measures of features (e.g., traffic volume, accident counts) between locations as their correlation strengths and pre-compute a **time invariant** adjacency matrix of locations [41]. In fact, studies [10][24] demonstrate that the influence of traffic conditions on events varies over different periods. Inspired by a related work [39] on a different problem, we use historical events to generate a static graph and learn a time-variant graph from $F_{ST}$ (e.g., traffic volume) in each time interval. Intuitively, we use $F_T$ (e.g. hour of the day) to learn the weights of dynamic graph vs. static graph to be considered in the graph convolution. The key equations of generating the adaptive graph adjacency are shown below:

$$Z_1 = \tanh(\alpha E_1 W_1) \tag{4}$$

$$Z_2 = \tanh(\alpha E_2 W_2) \tag{5}$$

$$\mathcal{A}_{dynamic} = \text{ReLU}(\tanh(\alpha(Z_1 Z_2^T - Z_2 Z_1^T))) \tag{6}$$

$$\beta = \text{sigmoid}(F_T W_3) \tag{7}$$

$$\mathcal{A} = \beta \mathcal{A}_{dynamic} + (1 - \beta) \mathcal{A}_{static} \tag{8}$$

Where $W_1$, $W_2$, and $W_3$ are learnable parameters. $E_1$ and $E_2$ are randomly initialized node embeddings that can be learned during training. We represent those embeddings by the spatiotemporal features $F_{ST}$ (e.g., traffic volume) to reveal the underlying dynamic connections between nodes. The subtraction and ReLU activation function in Eq. (6) lead to the asymmetric property of $\mathcal{A}_{dynamic}$. $\alpha$ is a hyperparameter to control the saturation rate. $\mathcal{A}_{static}$ is a pre-calculated adjacency matrix before training the model by computing the Pearson correlation coefficient between the risk scores of locations (nodes), where the correlation between node $i$ and node $j$, $a_{ij} = \frac{\sum_t (y_i - \bar{y_i})(y_j - \bar{y_j})}{\sqrt{\sum_t (y_i - \bar{y_i})^2 (y_j - \bar{y_j})^2}}$. The final adjacency matrix $\mathcal{A}$ is the weighted sum of $\mathcal{A}_{dynamic}$ and $\mathcal{A}_{static}$, and the weight $\beta$ is learned by a sigmoid activation function in Eq. 7 from the linear transformation of temporal feature $F_T$. Intuitively, the static adjacency matrix indicates a baseline correlation between different locations and it is pre-computed before training. This is also what most of the related work has been done. However, inspired by many observations from related studies [10][40], it is evident that we think this correlation is not always constant. Therefore, we use locations' time-variant features to construct a new adjacency matrix, and this dynamic adjacency matrix varies with time. We learn the parameters in this dynamic matrix during the network training process. Finally, we combine the static and the learned dynamic matrices through a learned weight $\beta$. In this way, a combined adjacency matrix can be treated as an adaptation from a static adjacency matrix considering the influence of other features during different periods. Finally, extracted embeddings are fed into LSTM layers and make final predictions of the risk scores $y$ for each location through a fully connected layer.

## 3.2 Local Ranking Metric and Hybrid Loss function

As previously mentioned, ranking-based metrics such as NDCG are not designed to handle spatial correlations. According to the first law of geography [35], nearby locations may share very similar socio-environmental attributes, thus it is challenging to rank neighboring locations correctly. However, the locations nearby with uncertain event patterns are worthy to be focused on so that more potential events can be discovered. Moreover, using NDCG on event ranking problems can cause over-concentrating on top-ranked locations and sacrificing prediction accuracy on other locations due to lower priority. To address those issues, we design a novel local ranking measurement named Local Normalized Discounted Cumulative Gain (L-NDCG) to measure spatially local ranking quality over every sub-region of the study area. The L-NDCG is calculated as:

$$\text{L-NDCG} = \frac{1}{T|\mathcal{S}_t|} \sum_{t=1}^{T} \sum_{s^t \in \mathcal{S}_t} \sum_{\hat{s} \in \mathcal{N}(s^t)} \frac{2^{y_{\hat{s}}^t} - 1}{Z_{\mathcal{N}(s^t)}^t \log_2(r(\hat{s}, \mathcal{N}(s^t)) + 1)}. \tag{9}$$

We use the same terminologies from Eq. 3. The unique part is that we compute an NDCG score for every location in the study area based on the local ranking of a subset of locations $\mathcal{N}(s^t)$, where $\mathcal{N}$ is a neighborhood of location $s^t$. We define the $\mathcal{N}$ as the Euclidean distance of coordinates smaller than $R$ in this work. $\mathcal{N}$ can be defined in other ways depending on the need of the problem. Essentially, L-NDCG is the average of NDCG scores for all subsets of locations. In this way, a few stationed hot-spot locations only take considerably large weight in their own NDCG scores and cannot over-influence the overall L-NDCG score. L-NDCG emphasizes ranking correctly on each subset of locations, and a greater L-NDCG score indicates that relatively more important locations can be distinguished from their nearby locations in a small region. Similar to optimizing NDCG, the ranking operator of L-NDCG is non-differentiable, thus we optimize its surrogates instead.

$$\max_{\mathbf{w} \in \mathbb{R}^d} L(\mathbf{w}) := \frac{1}{T|\mathcal{S}_t^+|} \sum_{t=1}^{T} \sum_{s^t \in \mathcal{S}_t^+} \sum_{\hat{s} \in \mathcal{N}(s^t)} \frac{2^{y_{\hat{s}}^t} - 1}{Z_{\mathcal{N}(s^t)}^t \log_2(\bar{g}(\mathbf{w}; \mathbf{x}_s^t, \mathcal{N}(s^t)) + 1)}. \tag{10}$$

Where $g(\mathbf{w}; \mathbf{x}_{\hat{s}}^t, \mathcal{N}(s^t))$ is a surrogate loss function similar to Eq. 2, and $\mathcal{N}(s^t)$ is a set of locations. Finally, to learn a trade-off between locally ranking quality and globally ranking quality we design a hybrid objective function consisting of both NDCG and L-NDCG

$$Loss = (1 - \sigma)\text{NDCG} + \sigma\text{L-NDCG} \tag{11}$$

Where $\sigma$ is a hyperparameter controlling the preference between NDCG and L-NDCG.

### 3.3  Importance-based Location Sampling with Spatial Filtering

To bridge the gap between capturing spatial correlations in a list of locations and optimizing the quality of predicted ranking, we propose a novel importance-based location sampling strategy with spatial filtering. Specifically, we first design an importance measure function to assign higher weights to locations with larger errors in predictions and higher ranking priority. Secondly, we sample locations based on the weights assigned by importance-measuring, so that important locations have a higher probability to be sampled. Afterward, only losses from sampled locations will be calculated in the objective function and considered during the optimization. In this way, the model pays attention to more important locations. Thirdly, we adjusted the importance scores every epoch so that the model learns to focus on different locations adaptively. Lastly, spatial filters are applied to smooth the importance scores in each epoch to achieve spatial-aware sampling. This allows nearby locations to be sampled in the same batch with high probability to help the model learn how to rank them. The training process is shown in algorithm 1

---

**Algorithm 1:** SpatialRank Training

---

**Input:** feature tensor $F$, event risk scores $y$, hyperparameter $\sigma$, standard deviation $\lambda$
**Output:** learned model $f_\theta$

1 Initialize probability set $P = \{p_1, p_2, ..., p_l\} \in L$;
2 **for** *each epoch* **do**
3   Compute $\hat{y} = f_\theta(x)$
4   Compute $loss_{NDCG} = \textbf{WeightedLoss}(y, \hat{y}, P)$
5   Compute $loss_{local} = \textbf{LocalWeightedLoss}(y, \hat{y}, P)$
6   $loss_{hybrid} = (1 - \sigma) \cdot loss_{NDCG} + \sigma \cdot loss_{local}$
7   **for** $s \in \mathcal{S}$ **do**
8    $E_s = \frac{1}{T} \sum_t \frac{2^{|y_s^t - \hat{y_s^t}|} - 1}{log_2(1 + r(y_s^t))}$
9   $E_{s'}^* = \sum_s \frac{E_s}{2\pi\lambda^2} e^{-\frac{x^2}{2\lambda^2}}$ for $s \in \mathcal{S}, x = dist(s', s)$
10   Update P = $\textbf{normorlize}(E')$
11   Compute gradient $\nabla f(\theta)$ by $loss_{hybrid}$
12   Update $\theta$ by $\nabla f(\theta)$
13 **return** $f_\theta$

---

Algorithm 1 shows the details of the importance-based sampling mechanism. The inputs include feature tensor $F$, event risk scores $y$, hyperparameter $\sigma$, and Gaussian standard deviation $\lambda$. The output is a learned model. The algorithm starts with initializing a set of equal-importance scores, which means the probabilities of locations being sampled are equal in the first epoch. Line 3 is forward propagation with current model parameters. Line 4 calculates the weighted NDCG losses given true label $y$, predicted labels $\hat{y}$, and current importance scores for locations. $Loss_{NDCG}$ is computed by Eq. 9, where importance scores $\{p_1, p_2, ..., p_s\} \in \mathcal{S}$ are normalized and treated as weights. Line 6 is our novel hybrid loss function discussed in Eq 10 to leverage the local ranking quality. The key step is to update the importance scores set based on current prediction errors and true ranking in Lines 7-8. We design a score function shown in Line 8 to leverage the errors made in predictions and the priority of true ranking for each location.

$$E_s = \frac{1}{T} \sum_t \frac{2^{|y_s^t - \hat{y_s^t}|} - 1}{log_2(1 + r(y_s^t))} \tag{12}$$

where $r()$ denotes a ranking function of the $i$-th location in the study area, and $|y_s^t - \hat{y_s^t}|$ is the absolute difference between predicted injuries and true injuries. Basically, a higher score $E_l$ indicates that

there are larger prediction errors and higher true ranking in this location $s$. Note that smaller $r(y_s^t)$ denotes a higher true ranking. To capture their geographical connections, we map the list of locations to their original locations on the grids, and then we apply a Gaussian filter to smooth the score distribution spatially in Line 9. $\lambda$ is the standard deviation. In this way, the model can learn better spatially correlated patterns, and avoid over-focusing on a few standalone locations but ignoring the the area nearby. Next, the smoothed scores $E^*$ are normalized into $P$, where $\sum_s P_s = 1$. Finally, we compute the gradient $\nabla f(\theta)$ by $loss_{hybrid}$ and update model parameters $\theta$. After a few iterations, we obtain the learned model $f_\theta$.

## 3.4 Complexity Analysis

In each iteration complexity of SpatialRank, we need to conduct forward propagation $h_t(s; \mathbf{w}), \forall s \in \mathcal{S}_t^+ \cup \mathcal{S}_t$ and back-propagation for computing $\nabla h_t(s; \mathbf{w}), \forall s \in \mathcal{S}_t^+ \cup \mathcal{S}_t$. The complexity of forwarding is $S = \sum_{t \in T}(|\mathcal{S}_t^+||\mathcal{N}(s) + |\mathcal{S}_t|)d \leq O(TSd)$, where $d$ is the model size. To compute $local_{NDCG}$ part, an extra cost on neighbor querying is needed to replace location list size on its complexity. The cost of computing $\bar{g}(\mathbf{w}; \mathbf{x}, \mathcal{S}_t)$ is $\sum_{t \in T}|\mathcal{S}_t^+||\mathcal{S}_t| \leq O(TS^2)$ for NDCG and $\sum_{t \in T}|\mathcal{S}_t^+||\mathcal{N}^2(s) \leq O(TS\mathcal{N}^2(s))$ for local NDCG. The size of $\mathcal{N}^2(s)$ is small, therefore the total cost is reduced to $O(TSd + TS^2)$.

# 4   Experiments

We perform comprehensive experiments on three real-world traffic accident and crime datasets from Chicago[2] and the State of Iowa[3]. Experiment results show that our proposed approach substantially outperforms state-of-art baselines on three datasets by up to $12.7\%$, $7.8\%$, and $4.2\%$ in NDCG respectively. The case study and results on the state of Iowa are shown in Appendix C.

**Data.** In the Chicago dataset, we collect data from the year 2019 to the year 2021. The first 18 months of this period are used as the training set, and the last 6 months of 2020 are used as the validating set. The year 2021 is used as a testing set. The area of Chicago is partitioned by 500 m $\times$ 500 m square cells and converted to a grid with the size of $64 \times 80$. For the crime dataset, we use total crimes as the risk score. For accident datasets, we use the number of injuries as the risk score. w **Baselines.** First, we use daily **Historical Average (HA)**. Next, we consider popular machine-learning methods including **Long Short-term Memory (LSTM)** [16], and **Convolutional LSTM Network (ConvLSTM)** [31]. Thirdly, we compare with recent methods such as **GSNet** [37], **Hetero-ConvLSTM**[40], and **HintNet** [5]. Moreover, we compared our optimization approach with other NDCG optimization solutions including **Cross Entropy (CE)**, **ApproxNDCG** [28], and **SONG** [29]. The details of the baselines are described in Appendix C.

**Metrics.** To measure the ranking quality of foremost locations, we test $K \in [30, 40, 50]$ on the test data. We use the metrics including NDCG, L-NDCG, and top-K precision (Prec)[23]. We report the average performance and standard deviation over 3 runs for three datasets.

## 4.1   Performance Comparison

In table 1, we can observe that our SpatialRank significantly outperforms other compared baselines in both datasets. We observe a similar trend among all metrics. On both datasets, Hetero-ConvLSTM and HintNet achieve similar results and outperform general machine learning methods such as LSTM and ConvLSTM. GSNet is designed on a dataset with a smaller grid size, thus performing worse on our larger grid. To study the effectiveness of learning the appropriate graph, we set the learning parameter $\beta$ as a fixed ratio of 0.5 in SpatialRank$^\#$. Oppositely, $\beta$ is a parameter to be learned based on temporal features in SpatialRank. In the Table 1 and Table. 2, each * indicates that the performance improvement of the proposed method over this baseline is statistically significant based on the student t-test with $\alpha = 0.05$ over three runs. The results show that the design of the time-aware graph convolution is able to improve performance and capture more dynamic variations in the graph, therefore performing better over different top-k rankings.

---

[2] https://data.cityofchicago.org/Transportation/Traffic-Crashes-Crashes/85ca-t3if

[3] https://icat.iowadot.gov/

## Table 1: Performance Comparison

| **Chicago Accident** | K=30 | | | K=40 | | | K=50 | | |
|---|---|---|---|---|---|---|---|---|---|
| | NDCG | Prec | L-NDCG | NDCG | Prec | L-NDCG | NDCG | Prec | L-NDCG |
| HA | .214±0 | .332±0 | .502±0 | .225±0 | .322±0 | .497±0 | .235±0 | .316±0 | .493±0 |
| LSTM | .215±2‰* | .392±2‰* | .519±3‰* | .225±1‰* | .380±2‰* | .543±3‰* | .249±1‰* | .368±2‰* | .544±3‰* |
| ConvLSTM | .225±5‰* | .410±4‰* | .558±3‰* | .236±1‰* | .388±4‰* | .563±8‰* | .252±1‰* | .366±2‰* | .540±8‰* |
| GSNet | .194±1‰* | .371±2‰* | .493±5‰* | .201±2‰* | .371±2‰* | .517±5‰* | .231±1‰* | .337±3‰* | .499±3‰* |
| Hetero-ConvLSTM | .229±2‰* | .401±1‰* | .557±3‰* | .240±1‰* | .395±4‰* | .564±3‰* | .255±3‰* | .375±2‰* | .551±3‰* |
| HintNet | .228±1‰* | .400±2‰* | .555±3‰* | .238±2‰* | .390±3‰* | .569±3‰* | .256±1‰* | .373±4‰* | .561±8‰* |
| SpatialRank [#] | .250±2‰ | .438±3‰ | .591±3‰ | .256±1‰ | .409±2‰ | .593±2‰ | .271±2‰ | .394±2‰ | .585±1‰ |
| SpatialRank | **.257**±1‰ | **.444**±3‰ | **.621**±4‰ | **.268**±1‰ | **.420**±1‰ | **.614**±2‰ | **.278**±3‰ | **.403**±1‰ | **.599**±1‰ |

| **Chicago Crime** | K=30 | | | K=40 | | | K=50 | | |
|---|---|---|---|---|---|---|---|---|---|
| | NDCG | Prec | L-NDCG | NDCG | Prec | L-NDCG | NDCG | Prec | L-NDCG |
| HA | .237±0 | .348±0 | .514±0 | .250±0 | .333±0 | .506±0 | .259±0 | .322±0 | .449±0 |
| LSTM | .246±1‰* | .327±2‰* | .517±3‰* | .257±1‰* | .329±2‰* | .521±3‰* | .262±3‰* | .314±5‰* | .512±3‰* |
| ConvLSTM | .313±2‰* | .415±4‰* | .617±4‰* | .325±2‰* | .404±1‰* | .607±6‰* | .333±2‰* | .387±4‰* | .599±3‰* |
| GSNet | .283±3‰* | .388±2‰* | .584±3‰* | .296±2‰* | .374±2‰* | .565±3‰* | .299±2‰* | .335±4‰* | .568±3‰* |
| Hetero-ConvLSTM | .346±3‰* | .468±3‰* | .657±4‰ | .365±1‰* | .452±3‰* | .642±6‰ | .374±4‰* | .433±4‰* | .638±4‰* |
| HintNet | .342±3‰* | .468±3‰* | .661±4‰ | .358±3‰* | .448±4‰* | .631±6‰* | .369±2‰* | .434±2‰* | .628±4‰* |
| SpatialRank [#] | .361±2‰ | .484±1‰ | **.670**±4‰ | .376±4‰ | .463±3‰ | **.655**±7‰ | .387±1‰ | **.446**±1‰ | **.651**±7‰ |
| SpatialRank | **.373**±2‰ | **.491**±3‰ | .665±4‰ | **.380**±2‰ | **.467**±5‰ | .647±6‰ | **.392**±2‰ | **.446**±3‰ | .644±6‰ |

[#] $\beta = 0.5$    ‰: $\times 10^{-3}$

## Table 2: Optimization Comparison

| **Chicago Accident** | K=30 | | | K=40 | | | K=50 | | |
|---|---|---|---|---|---|---|---|---|---|
| | NDCG | Prec | L-NDCG | NDCG | Prec | L-NDCG | NDCG | Prec | L-NDCG |
| CE | .232±1‰ | .424±1‰ | .588±2‰ | .253±1‰ | .415±2‰ | .588±2‰ | .266±1‰ | .400±1‰ | .580±2‰ |
| ApproxNDCG | .240±2‰ | .426±2‰ | .597±3‰ | .255±1‰ | .415±1‰ | .591±6‰ | .264±1‰ | .398±2‰ | .575±3‰ |
| SONG | .240±1‰ | .438±2‰ | .610±6‰ | .254±2‰ | .417±1‰ | .600±11‰ | .267±2‰ | .400±4‰ | .575±1‰ |
| SpatialRank | **.257**±1‰ | **.444**±3‰ | **.621**±4‰ | **.268**±1‰ | **.420**±1‰ | **.614**±2‰ | **.278**±3‰ | **.403**±1‰ | **.599**±1‰ |

| **Chicago Crime** | K=30 | | | K=40 | | | K=50 | | |
|---|---|---|---|---|---|---|---|---|---|
| | NDCG | Prec | L-NDCG | NDCG | Prec | L-NDCG | NDCG | Prec | L-NDCG |
| CE | .362±1‰ | .475±1‰ | .653±3‰ | .378±1‰ | .455±1‰ | .646±1‰ | .386±1‰ | .441±2‰ | .644±2‰ |
| ApproxNDCG | .344±4‰ | .455±3‰ | .637±10‰ | .353±1‰ | .435±2‰ | **.658**±5‰ | .365±3‰ | .421±4‰ | .619±6‰ |
| SONG | .364±2‰ | .484±3‰ | .660±5‰ | .379±4‰ | .466±3‰ | .456±1‰ | .390±2‰ | **.450**±3‰ | **.649**±6‰ |
| SpatialRank | **.373**±2‰ | **.491**±3‰ | **.665**±4‰ | **.380**±2‰ | **.467**±5‰ | .647±6‰ | **.392**±2‰ | .446±3‰ | .644±6‰ |

‰: $\times 10^{-3}$

### 4.2 Optimization Comparison

To evaluate the effectiveness of our proposed hybrid objective function and importance-based location sampling, we perform experiments on the same network architecture but with different optimization solutions including Cross Entropy (CE), ApproxNDCG, and SONG. The results are shown in table 2. Methods designed to optimize NDCG consistently perform better than Cross Entropy. SpatialRank substantially outperforms SONG and ApproxNDCG and made a noticeable improvement on L-NDCG as it is considered in the objective function.

### 4.3 Ablation Study

We examine the effects of tuning hyper-parameter $\sigma$ in the hybrid loss function. We present the results in table 3. Recall that $\sigma$ decides the ratio of L-NDCG takes in the objective function. $\sigma = 0$ means L-NDCG is not considered in the objective function. Starting from $\sigma = 0$, we observe a trend that the overall performance improves steadily while $\sigma$ goes up, and it reaches the best performance

TABLE 3: ABLATION STUDY

| CHICAGO ACCIDENT | K=30 | | | K=40 | | | K=50 | | |
| --- | --- | --- | --- | --- | --- | --- | --- | --- | --- |
| | NDCG | PREC | L-NDCG | NDCG | PREC | L-NDCG | NDCG | PREC | L-NDCG |
| $\sigma = 0.0$ | 0.251 | 0.439 | 0.610 | 0.263 | 0.407 | **0.612** | 0.274 | 0.289 | 0.605 |
| $\sigma = 0.05$ | 0.239 | 0.416 | 0.559 | 0.254 | 0.394 | 0.600 | 0.266 | 0.386 | 0.597 |
| $\sigma = 0.1$ | **0.256** | 0.443 | **0.621** | **0.268** | **0.421** | 0.608 | **0.276** | **0.400** | 0.599 |
| $\sigma = 0.2$ | 0.250 | **0.444** | 0.616 | 0.261 | 0.411 | 0.605 | 0.271 | 0.394 | **0.603** |
| $\sigma = 0.3$ | 0.234 | 0.422 | 0.606 | 0.243 | 0.388 | 0.59 | 0.253 | 0.374 | 0.591 |
| CHICAGO CRIME | K=30 | | | K=40 | | | K=50 | | |
| | NDCG | PREC | L-NDCG | NDCG | PREC | L-NDCG | NDCG | PREC | L-NDCG |
| $\sigma = 0.0$ | 0.366 | 0.489 | 0.661 | 0.378 | 0.464 | 0.644 | 0.385 | 0.445 | 0.642 |
| $\sigma = 0.05$ | 0.366 | 0.489 | 0.659 | 0.378 | **0.467** | **0.649** | 0.382 | 0.447 | **0.643** |
| $\sigma = 0.1$ | **0.371** | **0.494** | **0.665** | **0.383** | **0.467** | 0.644 | **0.391** | **0.450** | 0.642 |
| $\sigma = 0.2$ | 0.356 | 0.474 | 0.654 | 0.367 | 0.448 | 0.643 | 0.372 | 0.429 | 0.623 |
| $\sigma = 0.3$ | 0.345 | 0.457 | 0.658 | 0.361 | 0.445 | 0.641 | 0.373 | 0.431 | 0.631 |

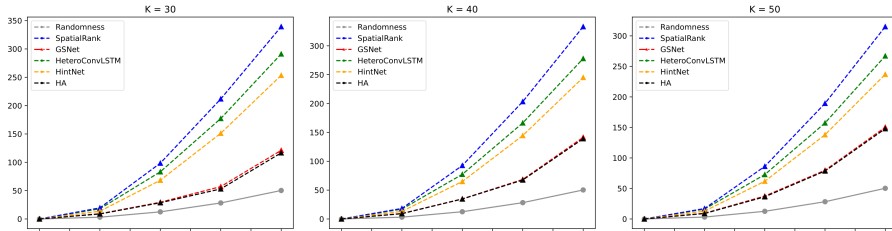

Figure 2: Comparison of cross-K function in Chicago Accident.

at $\sigma$ equals 0.1 in the accident and crime dataset. It indicates that considering a reasonable weight of L-NDCG in the objective function boosts overall performance. These results demonstrate the importance of considering local ranking because NDCG, L-NDCG, and precision can be all improved.

### 4.4 Cross-K function

We use the Cross-K function[14] with Monte Carlo Simulation[33] to evaluate the accuracy of predicted locations. The Cross-K function measures the spatial correlation between the predicted locations and true locations. Specifically, we calculate the average density of predictions within every distance $d$ of a true event in each day as shown in Eq. 13:

$$\hat{K}(d) = \lambda_j^{-1} \sum_{i \neq j} I(d_{ij} \leq d)/n, \tag{13}$$

where $\lambda$ is global density of event $j$, and $I()$ is an identity function which equals one if real distance $d_{ij}$ is smaller than $d$, else equals zero. $n$ is the number of events $i$. The results are shown in Figure 2. The grey curve represents the complete spatial randomness and we use it as a reference baseline. Higher is better. Our SpatialRank achieves the best predictions in both datasets, which indicates that the predictions of SpatialRank are significantly spatially correlated with ground truth. The similar results on the other two datasets are shown in Appendix C in the supplementary materials.

**Additional Results** and a **Case Study** to demonstrate successful prediction examples are included in the Supplementary materials.

# 5  Related Work

**Urban Event forecasting** has been widely studied in the past few decades. Most early studies [13] [21] [6][19] rely on small-scale and single-source datasets with limited types of features, thus the prediction accuracy is limited. Notably, Zhou et al. [42] proposed a differential time-varying graph convolution network capturing traffic changes and improving prediction accuracy. Similarly, Wang et al. [37] proposed GSNet with a geographical module and a weighted loss function to capture semantic spatial-temporal correlations among regions and solve data sparsity issues. Addressing the issue of spatial heterogeneity, Yuan et al. [40] proposed Hetero-ConvLSTM to leverage an ensemble of predictions from models learned from pre-selected sub-regions. Furthermore, An et al. [5] proposed HintNet partitions the study area hierarchically and transfers learned knowledge over different regions to improve performance. However, most existing models rely on optimizing cross-entropy and the objective is to make accurate predictions on every location. **Learning to rank** is an extensively studied area in recommendation systems and search engines [22]. NDCG is a widely adopted metric to measure ranking quality. The prominent class of methods involves approximating ranks in NDCG using smooth functions and subsequently optimizing the resultant surrogates. Taylor et al. [34] tries to use rank distributions to smooth NDCG, but suffers from high computational cost. Similarly, Qin et al. [28] approximates the indicator function by a generalized sigmoid function with a top-k variant. Noticeably, Qiu et al. [29] develop stochastic algorithms optimizing the surrogates for NDCG and its top-K variant. Although it is possible to directly apply the existing ranking methods to the urban event ranking problem, the performance tends to be unsatisfactory as these methods commonly assume independence between items and queries and lack the ability to handle spatiotemporal autocorrelation in the data.

# 6  Conclusion and Limitations

In this work, we formulate event forecasting as a location ranking problem. We propose a novel method SpatialRank to learn from spatiotemporal data by optimizing NDCG surrogates. To capture dynamic spatial correlations, we design an adaptive graph convolution layer to learn the graph from features. Furthermore, we propose a hybrid loss function to capture potential risks around hot-spot regions, and a novel ranking-based importance sampling mechanism to leverage the importance of each location considered during the model training. Extensive experimental results on three real-world datasets demonstrate the superiority of SpatialRank compared to baseline methods.

**Limitations**: the model's performance might be affected by other properties of data, such as spatial heterogeneity and sparsity. We observe less improvement over baselines on the Iowa dataset, partially due to that the data is sparser over a large area with heterogeneity. These are issues addressed by some of the prior work and can be addressed in our future work. In addition, the new algorithm increases the training time complexity due to the sampling steps. This is acceptable due to the relatively small number of locations and time periods in urban event datasets but may require extra work to generalize to large datasets.

# Acknowledgments and Disclosure of Funding

T. Yang was partially supported by NSF Career Award 18844403, NSF Fair AI Grant 2147253, NSF RI Grant 2110545.

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
