# OpenReview forum: "SpatialRank: Urban Event Ranking with NDCG Optimization on Spatiotemporal Data"
_NeurIPS.cc/2023/Conference — NeurIPS 2023 poster_

### Official Review · Reviewer_VBY2 · 2023-07-05

**Soundness:** 3 good
**Presentation:** 2 fair
**Contribution:** 3 good
**Rating:** 5
**Confidence:** 2

**Summary:**

This paper casts event forecasting as location ranking problem. They propose a spatial event ranking approach called SpatialRank. SpatialRank optimizes the NDCG metric while taking spatiotemporal autocorrelation into account.

Spatialrank uses a graph convolutional network to encode autocorrelated input and applies NDCG optimization on the top fully-connected layer. Based on the experiments on 3 datasets, it outperforms the underlying approaches on precision@k metric.

**Strengths:**

This paper is a good application of graph convolutional networks to autocorrelated spatiotemporal data to solve the event ranking problem. None of the individual pieces are unique, but the compound technique is a novel application.

* Introduction of L-NDCG is novel as it addresses the problem of hotspots dominating the overall NDCG metric.
* It is also nice that the authors performed an ablation study -- albeit being rather minimal.
* Ideas are fairly clear and explained well.
* The work might be significant if its applicability, which is teased in Section 6, is discussed in more detail.

**Weaknesses:**

A major weakness of the paper is its presentation. It needs attention to detail as well as how it is presented overall before being ready to be published. Some of the presentation issues are as follows:
* Lines 41-61 is basically a Related Work section. However, the reader is greeted with an actual Related Work section on the same page where most of the information is repeated.
* Line 151. What's Adam-style optimizer? Do you mean to say gradient-based optimizers?
* The main contribution of the paper is illustrated in Figure 1 of the Appendix. It deserves to be in the actual paper.
* Iowa dataset needs to be in the paper and the existing tables should be squeezed as they don't carry too much information to deserve that much space.
* The paper occasionally leads the reader to assume that the actual task is to increase the NDCG score although the problem is to predict top k events.
* A few spelling, capitalization, or other errors:
  * Line 86: Related Works -> Related Work
  * Line 182: Relu -> ReLU
  * Algorithm 1: Normorlize -> normalize
  * Line 280: We -> we
  * Line 281: Convlstm -> ConvLSTM
  * K@30 -- can't really tell this means. It is maybe K=30?

One other main weakness of the paper is its over-reliance on NDCG. First of all, experiments should't compare the approaches based on NDCG as it is the metric this approach is optimizing for as it wouldn't make a fair comparison per Goodhart’s law. Secondly, it is stated that "NDCG is not a perfect metric for our problems as it neither measures the local ranking quality nor considers spatial autocorrelation among locations." However, NDCG is still treated as somewhat ground truth throughout the paper.

Once the problem is converted into NDCG optimization, other approaches can also be utilized. For the experiments, tree-based ranking models can also be evaluated to replace the final FC layer as they have the potential to perform better.

**Questions:**

* What are the quantitative measures of performance wrt baselines?
* Do subregions need to be given?

**Limitations:**

Authors have addressed a few limitations including performance, data sparsity. To stress-test the approach the authors are advised to test their method on more variety of datasets. One can't confidently enumerate all limitations before it is tested on many datasets.

---

> ### Author Rebuttal · Authors · 2023-08-09
>
> Dear reviewer VBY2,
>
> Thank you very much for your comments! Below we answer your questions and address your comments.
>
> Q: The reader is greeted with an actual Related Work section on the same page where most of the information is repeated.
>
> A: Thank you for your suggestions! We will rephrase the related work section and reduce the repeated information in revision.
>
> ***
> Q: What's Adam-style optimizer? Do you mean to say gradient-based optimizers?
>
> A: Sorry for the confusion. We mean the Adam optimizer [1], which is an extension to stochastic gradient descent optimizer. Adam optimizer computes individual adaptive learning rates for different parameters from estimates of the first and second moments of the gradients. We found that Adam optimizer performs better in our datasets, so we use this term in the paper. Theoretically, equation 3 can be optimized by any gradient-based optimizer. To be accurate, we will use ‘gradient-based optimizers’ in our updated paper.
>
> ***
>
> Q: Figure 1 deserves to be in the actual paper.
>
> A: Thank you for the suggestion. Figure 1 will be moved to Section 3 in the revision.
>
> ***
>
> Q: Iowa dataset needs to be in the paper and the existing tables should be squeezed.
>
> A: Thank you for the suggestion. Given this cross-domain spatiotemporal ranking problem, we feel the need to use multiple baselines, metrics, and analysis methods from both learning-to-rank and spatiotemporal event forecasting domains to guarantee the completeness and consistency of the experiments. We agree with you that including results from a different geographic area will make the conclusion of the paper stronger. We will rearrange the materials to fit the results of the Iowa data. **Please check the attached pdf in the overall rebuttal**
> ***
> Q: The paper occasionally leads the reader to assume that the actual task is to increase the NDCG score although the problem is to predict top k events.
>
> A: To clarify, the goal is to predict the top k events with the highest risk scores. We use a hybrid NDCG score as the objective function to evaluate the ranking quality, where optimizing (maximizing) the hybrid NDCG will generally lead to a model that can better rank the top events. The traditional NDCG score is part of the objective function but simply maximizing itself might not give the best ranking results due to spatial autocorrelation of the data. This is the motivation for us to propose the hybrid NDCG loss with the local NDCG term and the new algorithm.
> ***
> Q: A few spelling, capitalization, or other errors.
>
> A: Thank you for pointing them out! We will correct all issues in the revision.
>
> ***
>
> Q: K@30? maybe K=30?
>
> A: Yes, it is correct. For example, NDCG K@30 means only top-30 ranked locations contribute to the final NDCG score. Details can be found in Experiments Section.
>
> ***
>
> Q: One other main weakness of the paper is its over-reliance on NDCG. experiments should't compare the approaches based on NDCG as it is the metric this approach is optimizing for as it wouldn't make a fair comparison per Goodhart’s law.
>
> A: Firstly, NDCG is only one of the three metrics used in our experiments. We also compared top-k precision and local NDCG. NDCG and top-k precision are the most widely accepted metrics in learning to rank problems. Secondly, our method does not directly optimize any of the individual metrics. It indeed optimizes a hybrid NDCG, which is the combination of NDCG and L-NDCG. By contrast, some of the SOTA baselines (such as SONG and ApproxNDCG) directly optimize NDCG as their objective. Therefore, the comparison is actually unfair to our model. However, SpatialRank still achieves superior performance in all the three metrics, including NDCG, compared to these methods. Lastly, the motivation of ranking locations based on importance is to meet the real-world demand that deploying limited law enforcement resources to the most needed places. Existing methods fail to leverage the foremost important locations and thus perform poorly in terms of ranking quality.
> ***
>
> Q: It is stated that "NDCG is not a perfect metric for our problems as it neither measures the local ranking quality nor considers spatial autocorrelation among locations." However, NDCG is still treated as somewhat ground truth throughout the pape?
>
> A: We believe NDCG is a reasonable measurement in this problem. Though we believe the proposed L-NDCG is more suitable for this problem, it is more convincing to also include existing widely accepted ranking measures such as NDCG and top-k precision. We are not using it as a ground truth. We use it as a building block of our new hybrid loss function, which addresses the limitations of the traditional NDCG measure.
> ***
>
>
> Q:  Do subregions need to be given?
>
> A: A subregion is defined as the spatial neighborhood of each location including itself. Users can define subregions based on their knowledge or common assumptions. In this paper we use a common choice of 3x3 grid cells around each location.
>
> ***
>
> Q: What are the quantitative measures of performance wrt baselines?
>
> A: We use four metrics to measure the performance of baselines and proposed method including NDCG, L-NDCG, top-K precision, and cross-k function:
> *  NDCG is a metric of ranking quality or the relevance of foremost important items, and a higher score indicates better ranking quality.
> *  L-NDCG is proposed in our paper and designed to measure spatially local ranking quality over every sub-region of the study area. Essentially, L-NDCG is the average of NDCG scores for all subsets of locations.
> *  Top-k precision is a widely accepted measurement in learning-to-rank problems. In our cases, it is equal to the number of locations in top-k recommendations where events occurred divided by the number of recommendations k
> *  The Cross-K function measures the spatial correlation between the predicted locations and true locations. Details can be found in Section 4.4
>
> ***
>
> [1] Kingma, et al.. Adam: A Method for Stochastic Optimization. 2014

---

> > ### Comment · Reviewer_VBY2 · 2023-08-15
> > **Recognition of authors' rebuttal**
> >
> > Thank you for providing responses to my questions.
> >
> >  * Apologies for not being clear with my question about "What are the quantitative measures of performance wrt baselines?" I meant the runtime performance. Roughly, how long does it take to train and do inference compared to others?
> >  * I agree with the rationale for using NDCG and coming up with L-NDCG, but not convinced by "Secondly, our method does not directly optimize any of the individual metrics. It indeed optimizes a hybrid NDCG, which is the combination of NDCG and L-NDCG." It still optimizes for NDCG and/or L-NDCG directly (modulo surrogate versions) -- making a linear combination of the two does not invalidate the fact that model optimizes the individual terms. Consider a loss function with cross entropy loss on labels and L2 weights as a regularization term. Final loss is $CE + W^2$. It doesn't change the fact that the model is optimizing on the labels.
> >  * For my second point about NDCG, the paper starts with "The problem of urban event ranking aims at predicting the top-k most risky locations of future events such as traffic accidents and crimes." It is argued in the paper that NDCG is not a good measure for this problem ("NDCG is not a perfect metric for our problems ..."). Moreover, NCDG is one of the three metrics in the final evaluation. These points don't make a consistent narrative as NDCG is not necessitated by the underlying problem but imposed by the authors to begin with.
> >  * Sorry for insisting on this nitpicking comment: The table should have NDCG@K, Prec@K, and L-NDCG@K and column group headings should be K=30, K=40, K=50.

---

> > > ### Author Response · Authors · 2023-08-17
> > >
> > >
> > >
> > > Dear reviewer VBY2
> > >
> > > **Q**: Apologies for not being clear with my question about "What are the quantitative measures of performance wrt baselines?" I meant the runtime performance. Roughly, how long does it take to train and do inference compared to others?
> > >
> > > **A**: Thank you for the suggestion! We have added comparisons with SOTA methods on average training time in seconds per epoch and inference time on testing dataset. The results are copied below and will be added to the revision. The Chicago crime dataset and the Chicago accident dataset have the same input feature; thus, have equivalent training costs. In summary, SpatialRank trains faster than two SOTA baselines HintNet and GSNet on both datasets. It is only slower than HeteroConvLSTM but the training times of the two are on the same order of magnitude. The training phase of SpatialRank is slow because of computing nested L-NDCG loss function.  Without extra cost on proposed optimization techniques, SpatialRank is significantly faster than all baselines in the inference phase. Given the improvement in prediction performance, we believe the cost of training time is acceptable, which will not affect the predicting efficiency of the proposed method.
> > >
> > > | Training Time| SpatialRank | HintNet | HeteroConvLSTM|GSNet|
> > > | ----------- | ----------- |----------- | ----------- | ----------- |
> > > | Chicago  | 88.2 | 132.1 | 47.7 | 98.8 |
> > > | Iowa | 76.5 | 117.5 | 41.6 | 83.5 |
> > >
> > > | Inference Time | SpatialRank | HintNet | HeteroConvLSTM|GSNet |
> > > | ----------- | ----------- |----------- | ----------- | ----------- |
> > > | Chicago| 5.6|51.2|14.4|21.1|
> > > | Iowa| 5.1|41.7| 12.3| 16.2|
> > >
> > >
> > > **Q**: I agree with the rationale for using NDCG and coming up with L-NDCG, but not convinced by "Secondly, our method does not directly optimize any of the individual metrics. It indeed optimizes a hybrid NDCG, which is the combination of NDCG and L-NDCG." It still optimizes for NDCG and/or L-NDCG directly (modulo surrogate versions) -- making a linear combination of the two does not invalidate the fact that model optimizes the individual terms. Consider a loss function with cross-entropy loss on labels and L2 weights as a regularization term. Final loss is $CE + W^2$. It doesn't change the fact that the model is optimizing on the labels.
> > >
> > > **A**: We want to thank you for your feedback on our response. We are happy that you agree with the rationale of using NDCG and L-NDCG in this problem! Apologize for the confusing words used in our rebuttal. Yes, we agree with your comments that our proposed model optimizes for a linear combination of NDCG and/or L-NDCG directly. We want to clarify that the used term “not directly” in rebuttal means we don’t solely consider one of them but both of them at the same time. Some related work (e.g., SONG and ApproxNDCG) use NDCG as the sole objective to optimize. But our method still achieves a better NDCG compared to them. This means our method is more suitable than SOTA methods for solving our problem.  The used term ‘not directly’ is ambiguous to some extent. Fortunately, we don’t use this term to describe the proposed method in our paper. To be precise, we will emphasize that our model optimizes a linear combination of NDCG and L-NDCG in the revision. Thanks again for your suggestions!
> > >
> > >
> > > **Q**: For my second point about NDCG, the paper starts with "The problem of urban event ranking aims at predicting the top-k most risky locations of future events such as traffic accidents and crimes." It is argued in the paper that NDCG is not a good measure for this problem ("NDCG is not a perfect metric for our problems ..."). Moreover, NCDG is one of the three metrics in the final evaluation. These points don't make a consistent narrative as NDCG is not necessitated by the underlying problem but imposed by the authors to begin with.
> > >
> > > **A** Sorry for the confusion. We will clarify this sentence in our revision. We actually meant ‘NDCG is not a perfect objective function in our problem’ (e.g., used by those SOTA methods), but it is still a reasonable metric in our experiments. NDCG prioritizes the foremost important items in the ranking, which generally meets the need of solving our problem. To address the unique challenges of our problem, we propose the hybrid objective function which not only prioritizes foremost significant locations but also considers local rankings between neighbors. This is the novelty over existing NDCG measurement. However, as this paper tries to bridge the gap between urban event ranking problems and existing learning-to-rank problems such as recommendation systems, we believe it is necessary to consider this widely accepted ranking metric in our experiment.
> > >
> > >
> > > **Q**: Sorry for insisting on this nitpicking comment: The table should have NDCG@K, Prec@K, and L-NDCG@K and column group headings should be K=30, K=40, K=50.
> > >
> > > **A** Thank you for your suggestions. We will update those notations in our revision.
> > >
> > >
> > > Thanks again for your time reviewing our paper!!

---

### Official Review · Reviewer_JmsG · 2023-07-06

**Soundness:** 4 excellent
**Presentation:** 3 good
**Contribution:** 3 good
**Rating:** 7
**Confidence:** 4

**Summary:**

The paper proposes a method which applies learning to rank losses to spatiotemporal event prediction. The observed data is considered over a discrete spatial partitioning and considers a time series of feature information. Features are grouped into purely temporal, purely spatial (not-time dependent) and spatiotemporal information. In addition, there is a risk score >0 for time, location pairs where an event occurred. The examined task is now to predict the top-k riskiest location for the next time step.
The method encodes the state of the network using a know spatiotemporal prediction network which employs graph NN layers to encode the underlying spatial connectedness modelling a road network.
The paper's novelty is to learn a dynamic adjacency matrix from the spatiotemporal feature of each location, which allows us to dynamically model the current conditions, such as traffic.  This dynamic adjacency matrix is combined with the static one.
A second novelty is adapting the ranking score to consider local rankings, which replace the global rank of location the original NDCG with a ranking score within a local neighbourhood. The localized NDCG is also combined with global NDCG loss.
Fot training, the authors propose to compute a weight for each element which is based on the ranking score of the  prediction error.


**Strengths:**

The paper proposes a novel idea to compute a ranking of locations instead of directly computing an event probability or even the next event, which might be hard to measure. In other words, instead of predicting the event likelihood, a ranking is learnt, which should be easier to learn but sufficient for various tasks.

Though the general layout of the proposed method is to combine NDCG with a spatial prediction backbone, the paper adds three modifications and shows that they considerably improve the performance compared to baselines.

The paper is good to follow and provides a reproducible description of the proposed methods.

The experiments show improved results von two real-world data sets, and the authors provide an ablation study.

The paper provided sufficient supplementary material to check all the details.

**Weaknesses:**

Though I could follow the reasoning of computing a top-k query, the motivation why this information is enough in several applications could be motivated better in the introduction.

The ablation study seems only to compare various values of \delta. But it would also be interesting to see the impact of the other contributions. As the technical contribution seems to consist of three relatively independent adaptions to the spatiotemporal setting seeing how much each contribution added to the improvement might be very insightful.

The description of the dynamic adjacency matrix is a little bit cryptic. In particular, it is unclear how it is trained and why it should represent a dynamic adaption of the static adjacency matrix

**Questions:**

What is the intuition of subnetwork for the dynamic adjacency matrix, and what is it supposed to model? Can you give more details about the intuition of the architecture and how it naturally aligns with the static adjacency matrix?
(There should not be a link if there is no connection in the static matrix, right?).

Can you give details on how much the instance weighting in training helped to improve the performance or speed up the conference?

Similarly, is there an ablation study on the dynamic adjacency matrix?



**Limitations:**

The paper addresses its limitations in a dedicated paragraph.

---

> ### Author Rebuttal · Authors · 2023-08-09
>
> Dear reviewer JmsG,
>
> Thank you very much for your comments and appreciation of our paper! Below we address your questions and concerns.
>
> Q: Though I could follow the reasoning of computing a top-k query, the motivation why this information is enough in several applications could be motivated better in the introduction.
>
> A: Thanks for your suggestions! We will add extra explanations and citations on our motivation for formulating a ranking problem in the introduction. We believe being able to correctly rank the foremost important locations meets the real-world demand. Given limited law enforcement resources such as staffing, the most essential task is to ensure they are allocated to the riskiest areas. The value k can be adjusted by users and set larger to report more locations. Our case study in Figure 4 demonstrates that our proposed method can capture more hotspots than baselines who make predictions for all the locations.
> **Please also see more details in response #2 in overall rebuttal.**
>
> ***
> Q: The ablation study seems only to compare various values of \delta. But it would also be interesting to see the impact of the other contributions.
>
> A: Sorry for the confusion. Due to the page limit and massive experiment results, instead of having independent tables, we concatenate some of ablation studies into performance comparison and optimization comparison part. To study how adaptive convolution layer improves the performance, we set $\beta = 0.5$ in SpatialRank marked as SpatialRank* in performance comparison part in Table 1, so that ratio $\beta$ between static adjacency matrix and dynamic matrix is fixed. To study how our proposed optimization solution contributes to the performance, we compare our approach with other SOTA solutions on the same network architecture in the optimization comparison part in Table 2. All the results show that each of the three contributions of the paper plays an important role in the superior performance of SpatialRank.
>
> ***
>
> Q: It is unclear how it is trained and why it should represent a dynamic adaption of the static adjacency matrix. What is the intuition of subnetwork for the dynamic adjacency matrix, and what is it supposed to model? Can you give more details about the intuition of the architecture and how it naturally aligns with the static adjacency matrix? (There should not be a link if there is no connection in the static matrix, right?).
>
> A:  Dynamic adjacency matrix is learned from computing pairwise similarity between source node embeddings and target node embeddings. If nodes’ information is not available, $E_1$ and $E_2$ are randomly initialized node embeddings to be learned during training, we treat nodes’ $F_{ST}$ as embeddings in our case. Parameter $W_1$ and $W_2$ are learnable parameters. Intuitively, the static adjacency matrix indicates a baseline correlation between different locations. For example, event patterns in a residential area might be constantly correlated with patterns in a shopping center nearby. Static adjacency matrix is pre-computed before training. This is also what most of the related work has been done. However, inspired by many observations from related studies [1], it is evident that we think this correlation is not always constant. Therefore, we use locations’ time-variant features to construct a new adjacency matrix, and this dynamic adjacency matrix varies with time. We learn the parameters in this dynamic matrix during the network training process. Finally, we combine the static and the learned dynamic matrices through a learned weight (\beta). In this way, a combined adjacency matrix can be treated as adaptation from a static adjacency matrix considering the influence of other features during different periods. There should not be a link between two nodes in static matrix if the Pearson correlation coefficient between them is zero. The link between locations of the final adjacency matrix is determined by both the static and the dynamic graphs.
>
> ***
>
> Q:  Similarly, is there an ablation study on the dynamic adjacency matrix?
>
> A:  As we answered in the second question, there are ablation studies on the learned combined adjacency matrix versus fixed adjacency matrix in table 1. Results show that introducing the dynamic adjacency matrix can effectively improve the performance of the model (last two rows in Table 1). There could be other ways to construct the dynamic adjacency matrix, but we did not perform additional evaluations on these choices as this is not our main focus of the paper.
> ***
>
> Q:  Can you give details on how much the instance weighting in training helped to improve the performance or speed up the conference?
>
> A:  Thanks for pointing this out. We will add an extra ablation study to the camera-ready paper and copied below. We compare SpatialRank to the same model with equal weights. The results show that instance weighting constantly improves the prediction performance.
>
> | Dataset | methods| NDCG@30 | PREC@30 | L-NDCG@30 | NDCG@40 | PREC@40 | L-NDCG@40 | NDCG@50 | PREC@50 | L-NDCG@50 |
> | ----------- | ----------- | ----------- |----------- | ----------- | ----------- |----------- | ----------- |----------- |----------- | ----------- |
> | Chicago Accident | equal-weights  | .255      | .441 | .622 | .265      | .417 | .613 |.274      | .401 | .595 |
> | Chicago Accident    |  SpatialRank  | .257      | .444 | .621 | .268      | .420 | .614 |.278      | .403 | .599 |
> | Chicago Crime  |  equal -weights    | .373      | .480 | .660 | .379      | .466 | .456 |.390      | .450 | .649 |
> | Chicago Crime |  SpatialRank  | .373      | .491 | .665 | .380      | .467 | .647 |.392      | .446 | .644 |
> | Iowa  |  equal -weights  | .531      | .304 | .617 | .557      | .264 | .591 |.573      | .225 | .546 |
> | Iowa  |  SpatialRank  | .540      | .309 | .618 | .563      | .268 | .600 |.585      | .232 | .550 |
>
> ***
>
>
> [1] Carey et al “Impact of Daylight Saving Time on Road Traffic Collision Risk: a Systematic Review.”

---

> > ### Comment · Reviewer_JmsG · 2023-08-13
> > **recognition of author rebuttal**
> >
> > Thank you for carefully addressing the brought-up points about your submission. I also appreciate the extra effort in providing the additional experimental results.

---

> > > ### Author Response · Authors · 2023-08-14
> > >
> > > Reviewer JmsG,
> > >
> > > we appreciate your feedback and suggestion to our paper. We are glad the paper was improved.
> > >
> > > Thank you very much, Reviewer JmsG!

---

### Official Review · Reviewer_yV4P · 2023-07-06

**Soundness:** 2 fair
**Presentation:** 1 poor
**Contribution:** 2 fair
**Rating:** 5
**Confidence:** 4

**Summary:**

The paper proposes a deep learning model called SpatialRank that predicts the top-k riskiest locations of future events such as traffic accidents and crimes by optimizing a spatial version of the NDCG measure. The model features adaptive graph convolution layers that learn the spatiotemporal dependencies from data, a hybrid loss function that balances global and local ranking quality, and an importance-sampling with spatial filtering algorithm that guides the model to focus on important locations. The model is evaluated on three real-world datasets from Chicago and Iowa, and outperforms several methods in terms of NDCG, L-NDCG, and precision. The model also demonstrates better spatial correlation with ground truth using the cross-K function.

**Strengths:**

S1 The proposed method is evaluated on three different real-life datasets.

S2 The idea of introducing the NDCG metric into a location-based learning problem is somewhat interesting.

**Weaknesses:**

Though this paper arises an interesting problem, I have the following major concerns about this paper:

(1) The motivation for urban event ranking is weak. The paper does not highlight the significance and novelty of this problem, and why it is more important than making event predictions for each location. The paper should provide more evidence or examples to motivate the need and value of ranking locations for future events.

 (2) The technical contribution is limited and the model is not novel enough. The paper does not clearly state how the proposed model differs from existing methods in terms of problem formulation, model design, optimization strategy, or evaluation metric. The designed NDCG loss function is quite similar to Eq. (3). I do not think there is much difference between these two functions. The paper should provide more details and analysis to demonstrate the advantages and challenges of the proposed approach.

(3) The model uses a Euclidean distance to define the neighborhood of each location, which may not reflect the actual spatial proximity or connectivity of locations in terms of road network or travel distance.

(4) The presentation and description of the framework is poor and incomplete. There are several unclear or confusing points in the paper, please refer to the above Questions.

(5) The experiments are not sufficient. There are several limitations or missing details in the experimental section, such as:
(5.1) No significance test to show whether the differences between SpatialRank and baselines are statistically significant or not.
(5.2) No ablation study experiments to show the effectiveness or necessity of each component of SpatialRank, such as the dynamic graph generation module or the designed NDCG loss function.

- Minor concerns:

(6) There are some grammar issues in the paper, such as “learning-to-ranking” in Line 51, which should be corrected.

(7) The paper lacks a framework figure to illustrate the overall architecture and workflow of the model, which makes it hard to follow and understand.

**Questions:**

Q1. Directly constructing from temporal events is more effective and explicit. Why generating a dynamic graph from features is necessary or effective? How does it capture the dynamic spatiotemporal dependencies among locations? How does it compare with directly constructing a graph from temporal events?

Q2. Why the method set E1 = E2 = FST in Eq. (4) and Eq. (5)? Since these two equations are almost same, why combine the two embedding Z_1 and Z_2 in Eq. (6) for generating the dynamic adjacency matrix?

**Limitations:**

The authors have discussed the limitations of their work.

---

> ### Author Rebuttal · Authors · 2023-08-09
>
> Dear reviewer yV4P
>
> Thank you very much for your comments! Please find our responses below:
>
> Q: The motivation for urban event ranking is weak. The paper should provide more evidence or examples to motivate the need and value of ranking locations for future events.
>
> A: We believe being able to correctly rank the foremost important locations meets the real-world demand because many related works and reports have shown that deploying limited staffing resources to the most needed places is a crucial task for law enforcements. Case studies also show that our method can find the riskiest locations and capture more hotspots than baselines who predict for all locations. **We provide more details in response #2 in the overall rebuttal.**
> * * *
> Q:  The technical contribution is limited and the model is not novel enough. The paper should provide more details and analysis to demonstrate the advantages and challenges of the proposed approach.
>
> A: We believe this paper presents important, novel, and valid contributions.
>
> First, this is the first paper to formulate an urban event forecasting problem as a spatial learning-to-rank problem and solve it by directly optimizing a spatial version of the NDCG measure.
>
> Second, we propose a novel local ranking measurement named L-NDCG and integrate it into our new loss function. This is to the best of our knowledge the first NDCG-based measure that considers spatial autocorrelation of data. The new loss **is different from the original NDCG** in not only an additional local ranking quality term, but also the underlying scientific assumptions and the non-trivial computational techniques needed to efficiently evaluate it.
>
> Third, we propose a novel importance-based location sampling algorithm to efficiently train the model to optimize the hybrid NDCG loss function. **The algorithm is also very different from traditional NDCG optimization techniques** as it uses spatial sampling to address the L-NDCG part of the loss function for the first time.
>
> In addition, we propose a network architecture with a novel adaptive convolution layer to capture the dynamic correlations between locations.
>
> Finally, we provide comprehensive experiments and evaluate the improvement gained through each of the above contributions.
>
> **More details in response #3 of overall rebuttal.**
>
> * * *
>
> Q:   The model uses a Euclidean distance to define the neighborhood of each location, which may not reflect the actual spatial proximity or connectivity of locations in terms of road network or travel distance.
>
> A:  We choose Euclidean distance because:
> * Euclidean distance is a widely accepted distance measure in the field of spatiotemporal forecasting problems, because it is computationally efficient and generally effective [1].
> * Computing actual spatial proximity in terms of if road network is computationally expensive and requires additional data, which might not be available to users.
> * The difference between using Euclidean distance and using travel distance is negligible in the small local neighborhood we consider. We computed the travel distance between the centroids of grid cells using road information, and found that the Euclidean distance is proportionally equivalent to the travel distance under current partition granularity in our Chicago and Iowa datasets. Even using Euclidean distance our model is already superior to SOTA methods.
> * * *
>
> Q:   There are some grammar issues in the paper.
>
> A:  Thank you for pointing this out. We will correct grammar issues.
>
> * * *
>
> Q:   The paper lacks a framework figure to illustrate the overall architecture and workflow of the model, which makes it hard to follow and understand.
>
> A:  Due to the limited paper space, the framework figure and extra experiment results were in supplementary material. We will improve the presentation in the revision.
>
> * * *
>
> Q:    Why the method set E1 = E2 = FST in Eq. (4) and Eq. (5)? Why combine the two embedding Z_1 and Z_2 in Eq. (6) for generating the dynamic adjacency matrix?
>
> A:   Sorry for the confusion. If nodes’ information is not available, $E_1$ and $E_2$ are randomly initialized node embeddings to be learned during training. In our case, we take advantage of two sets of $F_{ST}$ from the source node and target node respectively to represent node embeddings $E_1$ and $E_2$, so we are calculating the pairwise similarity between source node features and target node features in Eq. 6. The subtraction and ReLU activation function in Eq. (6) lead to the asymmetric property. In other words, we treat nodes' spatiotemporal features as embeddings to reveal underlying dynamic connections between nodes. We will rephrase these equations in the revision.
> * * *
>
> Q:    Directly constructing from temporal events is more effective and explicit. Why generating a dynamic graph from features is necessary or effective? How does it capture the dynamic spatiotemporal dependencies among locations? How does it compare with directly constructing a graph from temporal events?
>
> A:   Our proposed graph learning method learns graphs based on nodes’ features and dynamically updates graphs based on temporal information. As mentioned in Section 3, we use historical events to generate a static graph and learn a time-variant graph from $F_{ST}$ (e.g., traffic volume) through different periods. This design can capture the dynamic dependencies among locations in the real-world datasets. The experiment results in Table 1 further proves the benefits of considering dynamic graph versus static graphs. We are not very sure what you mean by constructing a graph from temporal events. The dataset covers a long time period (e.g., a few years). With events from each location during each day as nodes (~ millions), the graph, in particular the adjacency matrix could be too huge for any training algorithm.
>
> * * *
> [1] Zhe Jiang. 2018. A survey on spatial prediction methods. TKDE 31, 9 (2018), 1645–1664

---

> > ### Author Response · Authors · 2023-08-18
> >
> > Dear Reviewer yV4P,
> >
> > We sincerely appreciate your time reviewing our paper! We hope we have addressed all your concerns in our response. Please let us know if you have any additional questions.

---

> > ### Comment · Reviewer_yV4P · 2023-08-20
> > **Thanks for your careful response**
> >
> > Dear authors;
> >
> > Thanks for your careful response. I have raised my score. However, I still would like to give several suggestions for this paper:
> >
> > 1) An overall framework is necessary to show in the main manuscript to help the readers quickly get the general idea of the paper;
> >
> > 2) The presentation of the paper needs to be significantly improved before publication;
> >
> > 3) The improvement of the proposed method compared with baselines is incremental. Therefore, a significant test is deserved to prove the improvement is not trivial.
> >
> > Regards

---

> > > ### Author Response · Authors · 2023-08-21
> > >
> > > Thank you reviewer yV4P!
> > >
> > > We will make the following improvements in the revision regarding your suggestions:
> > >
> > > **Comment 1** An overall framework is necessary to show in the main manuscript to help the readers quickly get the general idea of the paper.
> > >
> > > **A:** Thanks for the suggestion! We will move the overall framework from supplementary material to the main manuscript and summarize the general idea of our paper at the beginning of the methodology section.
> > >
> > > **Comment 2** The presentation of the paper needs to be significantly improved before publication;
> > >
> > > **A:** Thanks for the suggestion! We will improve the presentation of our paper in the revision.
> > >
> > > **Comment 3** The improvement of the proposed method compared with baselines is incremental. Therefore, a significant test is deserved to prove the improvement is not trivial.
> > >
> > > **A:** Thanks for the suggestion! We provide the average performance with standard deviation over three runs in Table 1 and Table 2 of the updated pdf file in the author’s rebuttal. Furthermore, we have conducted the student t-test for these results, and the results demonstrate that our improvements over baselines are nontrivial. The results are copied below and will be updated in the revision. In the three tables, each * indicates that the performance improvement of the proposed method over this baseline is statistically significant based on the student t-test with $\alpha = 0.05$ over three runs.
> > >
> > > |Chicago Crime|NDCG@30|PREC@30|L-NDCG@30|NDCG@40|PREC@40|L-NDCG@40|NDCG@50|PREC@50|L-NDCG@50|
> > > |-----------| -----------|----------- | -----------|-----------|-----------| -----------|----------- |----------- | ----------- |
> > > | LSTM | .246+-0.001*| .327+-0.002*| .517+-0.003* | .257+-0.001*   | .329+-0.002*| .527+-0.003* |  .262+-0.003* |.314++0.005*    | .512+-0.003*|
> > > | ConvLSTM  | .313+-0.002*| .415+-0.004*| .617+-0.004* | .325+-0.002*   | .404+-0.001*| .607+-0.006* |  .333+-0.002* |.387++0.004*    | .599+-0.003*|
> > > | GSNet  | .283+-0.003*| .388+-0.002*| .584+-0.003* |.296+-0.002*   | .374+-0.002*| .565+-0.003* |  .299+-0.002* |.335++0.004*    | .568+-0.003*|
> > > | HeteroConvLSTM | .346+-0.003*| .468+-0.003*| .657+-0.004|.365+-0.001*   | .452+-0.004*| .642+-0.006|  .374+-0.004* |.433++0.004*    | .386+-0.004*|
> > > | HintNet| .342+-0.003*| .468+-0.003*| .661+-0.004 | .358+-0.003*   | .448+-0.004*| .631+-0.006*|  .369+-0.002* |.434++0.002*    | .628+-0.004*|
> > >
> > >
> > > |Chicago Accident|NDCG@30|PREC@30|L-NDCG@30|NDCG@40|PREC@40|L-NDCG@40|NDCG@50|PREC@50|L-NDCG@50|
> > > |-----------| -----------|----------- | -----------|-----------|-----------| -----------|----------- |----------- | ----------- |
> > > | LSTM | .215+-0.002*| .392+-0.002*| .519+-0.003* | .225+-0.002*   | .380+-0.002*| .543+-0.003* |  .249+-0.001* |.368++0.002*    | .544+-0.003*|
> > > | ConvLSTM  | .225+-0.005*| .410+-0.004*| .553+-0.003* | .236+-0.001*   | .388+-0.004*| .563+-0.008* |  .252+-0.001* |.366++0.002*    | .540+-0.008*|
> > > | GSNet  | .194+-0.001*| .371+-0.002*| .493+-0.005* |.201+-0.002*   | .371+-0.002*| .517+-0.005* |  .231+-0.001* |.337++0.004*    | .499+-0.003*|
> > > | HeteroConvLSTM | .229+-0.002*| .401+-0.001*| .557+-0.002* |.240+-0.001*   | .390+-0.004*| .564+-0.003* |  .225+-0.003* |.375++0.002*    | .551+-0.003*|
> > > | HintNet| .228+-0.001*| .400+-0.002*| .555+-0.003*| .238+-0.002*   | .390+-0.003*| .569+-0.003*|  .256+-0.001* |.373++0.004*    | .561+-0.008*|
> > >
> > > |Iowa|NDCG@30|PREC@30|L-NDCG@30|NDCG@40|PREC@40|L-NDCG@40|NDCG@50|PREC@50|L-NDCG@50|
> > > |-----------| -----------|----------- | -----------|-----------|-----------| -----------|----------- |----------- | ----------- |
> > > | LSTM | .503+-0.003*| .278+-0.003*| .573+-0.003* | .522+-0.001*   | .209+-0.001*| .518+-0.003* |  .519+-0.005* |.197++0.001*    | .474+-0.001*|
> > > | ConvLSTM  | .490+-0.003*| .282+-0.002*| .583+-0.001* | .507+-0.003*   | .207+-0.001*| .513+-0.004* |  .511+-0.003* |.189++0.003*    | .474+-0.008*|
> > > | GSNet  | .493+-0.002*| .265+-0.001*| .569+-0.003* | .509+-0.003*   | .222+-0.003*| .527+-0.003* |  .506+-0.005* |.207++0.001*    | .510+-0.003*|
> > > | HeteroConvLSTM | .518+-0.001*| .289+-0.002*| .617+-0.004 | .523+-0.005*   | .258+-0.001*| .589+-0.005* |  .543+-0.003* |.226++0.001*    | .534+-0.005*|
> > > | HintNet| .512+-0.005*| .289+-0.003*| .617+-0.001 | .542+-0.005*   | .243+-0.004*| .590+-0.009 |  .556+-0.003* |.209++0.002*    | .534+-0.008*|

---

### Official Review · Reviewer_RZQE · 2023-07-07

**Soundness:** 3 good
**Presentation:** 3 good
**Contribution:** 3 good
**Rating:** 8
**Confidence:** 4

**Summary:**

This paper investigates the problem of future urban event prediction on spatiotemporal data. This is an important problem for a broad range of urban application. Different from prior work, this paper for the first time predicts most likely future events by directly optimizing location ranking in the prediction through NDCG optimization. The authors propose a dynamic adjacency matrix of locations, a hybrid NDCG loss function and a spatial sampling algorithm to handle the unique challenges brought by spatiotemporal data. Experimental results on three datasets show that the proposed solution can beat state-of-the-art event prediction models as well as existing NDCG optimization solutions on crime and accident prediction.

**Strengths:**

1.	The problem investigated by the paper is significant for many urban applications such as crime prediction and accident forecasting.
2.	The paper is the first to use NDCG optimization on spatiotemporal data for location ranking. This is a novel idea for event prediction.
3.	The work also adds value to the literature of ranking algorithm for providing solutions on how to handle non-iid spatial data ranking.
4.	The experiments show that the proposed SpatialRank model outperforms not only event/accident prediction methods but also traditional NDCG optimization method. This suggests that the model is effective in addressing some of the unique challenges in spatiotemporal data.


**Weaknesses:**

1.	As discussed by the authors, the hybrid loss function might introduce significant computation cost increase to the learning algorithm. In addition to the complexity analysis, the authors should provide additional evidence (e.g., experiments) to justify the impact to training time.

2.	There are quite a lot of symbols in the paper and the authors should provide a table or summary of these symbols. Without such information, the complexity analysis part is a bit hard to follow.

3.	The experimental results should be presented with error bars.


**Questions:**

1.	What is the model architecture used for the optimization comparison experiment? How does the model architecture choice affect the performance of the optimizers? Is it possible that with a different network architecture the other optimizers such as SONG or approxNDCG can beat spatialRank?
2.	How is top-k precision defined in the experiments?


**Limitations:**

N/A.

---

> ### Author Rebuttal · Authors · 2023-08-09
>
> Dear reviewer RZQE,
>
> Thank you very much for your comments and appreciation of our paper! Below we address your questions and concerns.
>
> Q: In addition to the complexity analysis, the authors should provide additional evidence (e.g., experiments) to justify the impact to training time.
>
> A: Thank you for the suggestion! We have added comparisons with SOTA methods on average training time in seconds per epoch. The results are copied below and will be added to the revision. The Chicago crime dataset and the Chicago accident dataset have the same input feature; thus, have equivalent training costs. In summary, SpatialRank trains faster than two SOTA baselines HintNet and GSNet on both datasets. It is only slower than HeteroConvLSTM but the training times of the two are on the same order of magnitude. Given the improvement in prediction performance, we believe this is an acceptable cost, which will not affect the practical value of the proposed method.
> | Dataset      | SpatialRank | HintNet | HeteroConvLSTM | GSNet |
> | ----------- | ----------- |----------- | ----------- | ----------- |
> | Chicago     | 88.2      | 132.1 | 47.7 | 98.8 |
> | Iowa   | 76.5      | 117.5 | 41.6 | 83.5 |
>
> * * *
>
> Q: There are quite a lot of symbols in the paper and the authors should provide a table or summary of these symbols.
>
> A: Thank you for the suggestion. We will add a symbol table to the revision. Besides, explanations on symbols related to ranking optimization are included in Section 3.2 and Section 4.2. Symbols for problem formulation are introduced in Section 3.1.
> | Symbol | Explanations |
> | ----------- | ----------- |
> | $S$     | Spatial filed, study area |
> | $s$     | A partitioned location, grid cell|
> | $T$     | Temporal filed, study period|
> | $t$     | Time interval (e.g. hours, days)|
> | $F_T$     | Temporal features (e.g. weather, time)|
> | $F_S$     | Spatial features (e.g. POI) |
> | $F_{ST}$     | Spatiotemporal features (e.g. traffic conditions) |
> | $y$     | Event Risk Score|
> | $Z$     | Discounted Cumulative Gain (DCG) score|
> | $E$     | Node embedings|
> | $a$     | Pearson correlation coefficient|
> | $NDCG$     | Normalized Discounted Cumulative Gain|
> | $L-NDCG$     | Local Normalized Discounted Cumulative Gain|
> | $Prec$     | top-K precision|
> | $r()$     | Ranking function|
> | $N()$     | Neighbour querying|
>
>
>
> * * *
>
> Q: The experimental results should be presented with error bars
>
> A:  Thank you for pointing it out. We have revised the three experiment results and now report the average performance including NDCG, L-NDCG, and Precision with standard deviation over 3 runs in Table 1 and Table 2 of the updated pdf. The results show that SpatialRank significantly outperforms other compared baselines in three datasets. SpatialRank substantially outperforms SONG and ApproxNDCG and made a noticeable improvement on L-NDCG as it is considered in the objective function.
>
> * * *
>
> Q: What is the model architecture used for the optimization comparison experiment? How does the model architecture choice affect the performance of the optimizers? Is it possible that with a different network architecture the other optimizers such as SONG or approxNDCG can beat spatialRank?
>
> A:  The model architecture used for optimization comparison is described in Section 4.1, which utilizes LSTM to capture temporal dependencies and graph convolution layers to capture spatial dependencies. This is consistent with the model proposed in Section 3.1. . In the ablation study, we demonstrate that with the same network architecture but different loss functions and learning algorithms, SpatialRank can still beat other baselines (e.g., CE SONG). This proves that the improvements in performance came not solely from the network architecture proposed in 3.1 but all the three contributions. In fact, our proposed framework can improve the performance of other networks architectures as well, which is our core contribution.
>
> * * *
>
> Q: How is top-k precision defined in the experiments?
>
> A:  Top-k precision is a widely accepted measurement in learning-to-rank problems. In our cases, it is equal to the number of locations in top-k recommendations where events occurred divided by the number of recommendations k. It indicates how well our model can capture events within our recommendations. More explanations can be found in this paper Lu et al[1]. We will also cite this paper in the experiment section in the revision.
>
> * * *
>
> [1] Lu, Jing, et al. Sampling Wisely: Deep Image Embedding by Top-K Precision Optimization. IEEE, 2019

---

> > ### Comment · Reviewer_RZQE · 2023-08-17
> > **Comment**
> >
> > All of my concerns raised in the review have been addressed. The paper is well written with solid technical contributions and promising experiment result. I would like to increase my score.

---

> > > ### Author Response · Authors · 2023-08-18
> > >
> > > Thank you reviewer RZQE!

---

### Author Rebuttal · Authors · 2023-08-09

Q: The experimental results should be presented with error bars (RZQE)

A:  Thank you for pointing it out. We have revised the three experiment results and now report the average performance including NDCG, L-NDCG, and Precision with standard deviation over 3 runs in Table 1 and Table 2 of the updated pdf. The results show that SpatialRank significantly outperforms other compared baselines in three datasets. SpatialRank substantially outperforms SONG and ApproxNDCG and made a noticeable improvement on L-NDCG as it is considered in the objective function.

* * *

Q: Though I could follow the reasoning of computing a top-k query, the motivation why this information is enough in several applications could be motivated better in the introduction. (JmsG)

Q: The motivation for urban event ranking is weak. The paper does not highlight the significance and novelty of this problem, and why it is more important than making event predictions for each location. (yV4P)

A: Thanks for your suggestions! We will add extra explanations and citations on our motivation for formulating a ranking problem in the introduction of the revision. We believe being able to correctly rank the foremost important locations meets the real-world demand. Because deploying limited law enforcement resources to the most needed places is necessary. Chicago Police Department has utilized criminal intelligence analysis and data science techniques to help command staff determine where best to deploy resources [1]. According to Police Executive Research Forum, there were 42.7% more resignations among law enforcement but a 3.9% decrease in hiring new officers in 2021 compared to 2019 [2]. Meanwhile, the Federal Bureau of Investigation confirms that violent crime in 2020 has surged nearly 30% over 2019 [3]. Therefore, given such growth of crimes, deploying limited law enforcement resources to the most needed places is very necessary. In addition, our case study in Figure 4 of the supplementary material also demonstrates that our proposed method can prioritize the riskiest locations and capture more hotspots compared to baselines that make predictions for all the locations. This can potentially improve the deployment efficiency of police resources.

* * *

Q:  The technical contribution is limited and the model is not novel enough. The paper should provide more details and analysis to demonstrate the advantages and challenges of the proposed approach. (yV4P)

A: In this paper, we present the following contributions in terms of problem formulation, model design, optimization strategy, and evaluation metrics.
*  This is the first paper to formulate an urban event forecasting problem as a spatial learning-to-rank problem and solve it by directly optimizing a spatial version of the NDCG measure.  (As opposed to Existing works solve this problem by optimizing non-ranking-based metrics such as cross entropy by related work).
* We propose a novel local ranking measurement named L-NDCG and integrate it into our new loss function. This is to the best of our knowledge the first NDCG-based measure that considers spatial autocorrelation of data. The new hybrid loss **is quite different from the original NDCG** in not only an additional local ranking term, but also the underlying scientific assumptions as well as the non-trivial computational techniques needed to efficiently evaluate it. In Section 3.2, we explained the first law of geography [7] that nearby locations tend to be similar, which makes it challenging to rank neighboring locations correctly. L-NDCG emphasizes ranking correctly on each subset of locations so that important locations can be distinguished from their nearby locations. This is an important advancement over existing work.
*   We propose a novel importance-based location sampling algorithm to efficiently train the model to optimize the hybrid NDCG loss function, where we guide the model to pay more attention to important locations with higher training errors.  We explain more details in Algorithm 1 in the paper. **The algorithm is also very different from traditional NDCG optimization techniques** as it uses spatial sampling to address the L-NDCG part of the loss function for the first time.
* Our proposed optimization framework can work with different deep learning architectures. We proposed a variant of the existing graph neural networks with a novel adaptive convolution layer to capture the dynamic correlations between locations. Unlike prior works such as [4], our design allows us to dynamically capture the correlations between locations. This improves the performance of the method effectively.
* We provide comprehensive experiments on our proposed approach. We evaluate the performance of the adaptive convolution layers (SpatialRank* vs SpatialRank in Table 1), the advantages of L-NDCG in the hybrid objective function (ablation study on $\sigma$ in Table 3), and the effectiveness of our spatial sampling algorithm (compared with SOTA optimization methods in Table 2). The overall results clearly show that each of the above three contributions play an important role in helping SpatialRank outperform all the SOTA baselines on three different datasets. Therefore, we believe our paper has important, novel and valid contributions.

* * *

[1] [CPD Expands Smart Policing Technology to Support Strategic Deployment and CTA Safety]( https://home.chicagopolice.org/cpd-expands-smart-policing-technology-to-support-strategic-deployment-and-cta-safety/)

[2] [PERF survey shows steady staffing decrease over the past two years]( https://www.policeforum.org/workforcemarch2022)

[3] [US murder rate continued grim climb in 2021]( https://www.foxnews.com/us/us-murder-rate-continued-grim-climb-in-2021-new-fbi-estimates-show)

[4] Zhang, Yingxue, et al. TrafficGAN: Off-Deployment Traffic Estimation with Traffic Generative Adversarial Networks. IEEE, 2019, pp. 1474–79.

---

### Decision · Program_Chairs · 2023-09-21

**Decision:**

Accept (poster)

**Comment:**

The paper studies the important problem of ranking urban events and adopts a novel NDCG optimization on spatiotemporal data. The resulting SpatialRank algorithm shows strong experimental results. The reviewers appreciate the novelty of the method and the clarify in the presentation.